# Uric Acid Deteriorates Load-Free Cell Shortening of Cultured Adult Rat Ventricular Cardiomyocytes via Stimulation of Arginine Turnover

**DOI:** 10.3390/biology12010004

**Published:** 2022-12-20

**Authors:** Martin Weber, Rolf Schreckenberg, Klaus-Dieter Schlüter

**Affiliations:** Physiologisches Institut, Justus-Liebig-Universität, 35392 Gießen, Germany

**Keywords:** ornithine decarboxylase, calcium desensitization, arginine

## Abstract

**Simple Summary:**

Uric acid is a metabolite released by cells to detoxify themselves from nitrogen. Unlike other mammalians, hominids, and also birds, lack uricase, an enzyme that catalyzes the transformation of uric acid to allantoin. Therefore, high levels of uric acid accumulate in the plasma of hominids. Under certain conditions, i.e., the consumption of a fructose-rich diet, which is also often associated with diabetes, the concentrations of uric acid rise. Under such conditions, uric acid may in part exert effects on different cells. Using isolated and cultivated totally differentiated rat cardiomyocytes, we show that uric acid at plasma-relevant concentrations that are similar to those found in patients decreases the calcium-affinity of troponin in cardiomyocytes and thereby reduces cell function. Therefore, the accumulation of uric acid stresses cardiomyocytes, a mechanism that may contribute to heart failure under such conditions.

**Abstract:**

Hyperuricemia is a risk factor for heart disease. Cardiomyocytes produce uric acid via xanthine oxidase. The enzymatic reaction leads to oxidative stress in uric-acid-producing cells. However, extracellular uric acid is the largest scavenger of reactive oxygen species, specifically to nitrosative stress, which can directly affect cells. Here, the effect of plasma-relevant concentrations of uric acid on adult rat ventricular cardiomyocytes is analyzed. A concentration- and time-dependent reduction of load-free cell shortening is found. This is accompanied by an increased protein expression of ornithine decarboxylase, the rate-limiting enzyme of the polyamine metabolism, suggesting a higher arginine turnover. Subsequently, the effect of uric acid was attenuated if other arginine consumers, such as nitric oxide synthase, are blocked or arginine is added. In the presence of uric acid, calcium transients are increased in cardiomyocytes irrespective of the reduced cell shortening, indicating calcium desensitization. Supplementation of extracellular calcium or stimulation of intracellular calcium release by β-adrenergic receptor stimulation attenuates the uric-acid-dependent effect. The effects of uric acid are attenuated in the presence of a protein kinase C inhibitor, suggesting that the PKC-dependent phosphorylation of troponin triggers the desensitizing effect. In conclusion, high levels of uric acid stress cardiomyocytes by accelerating the arginine metabolism via the upregulation of ornithine decarboxylase.

## 1. Introduction

Uric acid (UA) is the end-product of the purine nucleotide catabolism and is enzymatically catalyzed by xanthine oxidase in a two-step reaction that converts hypoxanthine into xanthine and xanthine into UA. In both steps, superoxide is a byproduct. Superoxide can generate oxidative stress in cells specifically if the activity of xanthine oxidase is high or the activity of reactive-oxygen-species (ROS)-detoxifying enzymes is low. In most mammalians, UA can be transformed to allantoin by the enzyme uricase. However, hominids lack uricase and instead have a UA transport molecule in the kidney (URAT1) that allows the resorption of UA. Therefore, the UA plasma concentration in hominids is high compared to most mammalians. UA itself has antioxidative properties and it has been suggested that high plasma concentrations of UA in hominids contribute to its longevity compared to small rodents [1,2,3,4,5]. In line with this, mice that are haplo-deficit for uricase have an increased longevity [5]. However, under conditions of elevated plasma levels of UA, UA may become insoluble, specifically in areas with a low pH. The risk of high plasma levels of UA is also increased in patients consuming large amounts of fructose-rich diets, as it is often the case in people developing diabetes. In summary, high plasma concentrations of uric acid are considered as a risk factor for heart disease, although UA itself is antioxidative. Therefore, an understanding of the direct effects of UA on terminally differentiated cardiomyocytes is required to understand the pathophysiological role of UA in humans. 

The activity of xanthine oxidase is elevated in cardiomyocytes isolated from dogs with pacing-induced heart failure [6]. This is considered as an indication of oxidative stress in cardiomyocytes leading to heart failure. Indeed, several studies find an association between plasma UA and ischemic dilative cardiopathy, dilative cardiomyopathy, systolic dysfunction, and mitral valve dysfunction [7,8,9,10,11]. As a cut-off plasma level, UA concentrations above 8.0 or 8.7 mg/dL has been suggested [11,12]. On the mechanistic side, it is unclear why a molecule with antioxidative properties should be linked to cardiac disease. Even if stressed cardiomyocytes may produce more UA, the majority of UA in the plasma should still be derived from the liver and kidney. This point requires attention and directs the focus of the analysis to terminally differentiated adult cardiomyocytes. 

In this context, it is interesting that UA itself can affect cell function directly. UA can enhance PKC activity in endothelial cells [13], activate mitogen-activated protein kinase (MAPK) pathways in pancreatic cells [14], increase the expression of voltage-dependent potassium channels in atrial myocytes [15], and induce insulin resistance in cardiomyocytes [16], to name a few of cell-specific effects of UA. Therefore, it may be possible that UA also affects the contractile function of cardiomyocytes in a direct way, and thereby may be causally involved in the link between high plasma UA levels and heart failure.

To challenge this hypothesis, we expose isolated and cultured adult rat ventricular cardiomyocytes to plasma-relevant concentrations of UA. We use the load-free cell shortening of myocytes as an established surrogate parameter of the contractile function. The data of our study show that UA can increase the arginine metabolism in cardiomyocytes and exert effects on cardiomyocytes that are characteristic for arginine depletion. 

## 2. Materials and Methods

### 2.1. Ethical Concerns

The investigation conforms to the Guide for the Care and Use of Laboratory Animals published by the US National Institute of Health (NIH publication no. 85–23, revised 1996). The permission to scarify rats is registered at Justus-Liebig-University.

### 2.2. Materials

The calcium-sensing receptor protein was detected by the anticalcium sensing receptor antibody produced in rabbit (Sigma-Aldrich Chemie, Taufkirchen, Germany; SAB4503369M). Ornithine decarboxylase was detected by the anti-ODC antibody produced in goat (Santa-Cruz Biotechnology, Santa Cruz, CA, USA; F-14, sc-21516). The calcium-sensing receptor and ornithine decarboxylase levels were normalized to actin (rabbit antiactin antibody (Sigma-Aldrich Chemie, Taufkirchen, Germany; A2668)). Secondary antibodies were directed against rabbit IgG and coupled to alkaline phosphatase (Affinity Biologicals; Ancaster, ON, Canada; GAM-APHRP). SB202190, chelerythrine chloride, and SP600125 were purchased from Calbiochem (Merck Darmstadt, Darmstad, Germany). Uric acid, L-nitro arginine methyl ester (L-NAME), Nor-NOHA, and Difluoromethyl ornithine (DFMO) were purchased from Sigma-Aldrich Chemie, Taufkirchen, Germany. 

### 2.3. Cell Preparation and Cultivation

Adult rat ventricular cardiomyocytes were isolated from male 4-month-old Wistar rats, as described previously [17,18]. Briefly, the hearts were excised under deep isoflurane anesthesia, transferred rapidly into ice-cold saline, and mounted on the cannula of a Langendorff perfusion system. All subsequent steps were performed at 37 °C. First, the hearts were perfused with a calcium-free perfusion buffer for 5 min at 10 mL/min. Thereafter, perfusion was continued with recirculation but with the addition of 0.06% (*w*/*v*) crude collagenase and 25 µM CaCl_2_ at 5 mL/min. After 25 min, the ventricular tissue was minced and incubated for 20 min in a recirculating medium. The resulting cell suspension was filtered through a 200 µm nylon mesh. The filtered material was washed twice by centrifugation (3 min, 25 g) and resuspended in the collagenase-free perfusate, in which the calcium concentration was step-wise increased to 0.5 mmol/L. After a further centrifugation, the cell pellet was resuspended in serum-free culture medium (medium 199 with Earle’s salts, 5 mmol/L creatine, 2 mmol/L L-carnitine, 5 mmol/L taurine, 100 IU/mL penicillin, and 100 µg/mL streptomycin) and cells were plated on culture dishes (Falcon, type 3001). Medium change was performed after 2 h. Cells were analyzed thereafter or after 24 h incubation, as indicated in the Results section. The following inhibitors were used within this study: inhibition of p38 MAPK by SB202190 (10 µmol/L), inhibition of c-jun kinase by SP600125 (10 µmol/L), and inhibition of protein kinase C by chelerythrine chloride (CEC, 10 µmol/L). Concentrations were used according to references [19,20]. NO synthase inhibition was performed by L-nitro arginine methyl ester (L-NAME, 10 µmol/L) according to reference [21]. Ornithine decarboxylase was inhibited by difluoromethyl ornithine (DFMO, 100 µmol/L) as used before by reference [22]. Arginase was inhibited by administration of Nor-NOHA (100 µmol/L) according to reference [23].

### 2.4. Determination of Cell Contraction

Cells were allowed to contract at room temperature and analyzed using a cell-edge-detection system, as previously described [24]. Cells were stimulated via two AgCl electrodes with biphasic electrical stimuli of 0.5 ms duration. Each cell was stimulated at 2 Hz for 1 min. Every 15 s, contractions were recorded. The mean of these four measurements was used to define the cell shortening of a single cell. Nine cells were analyzed per experiment and the median of these cells was used at the shortening amplitude under the given conditions. Cells were analyzed in serum-free culture medium (see above) with or without further supplements, as indicated in the Results section. Data were expressed as the cell-shortening amplitude (µm) normalized to the diastolic cell length (µm) and expressed as % (ΔL/L). In addition, the maximal relaxation and maximal contraction velocity was analyzed. 

### 2.5. Quantification of Calcium Transients

Calcium transients were measured with the fluorescent dye fura-2 acetoxymethyl ester, as described before [25]. Adult rat ventricular cardiomyocytes were placed on glass cover slips and were loaded with fura-2 acetoxymethly ester (2.5 µmol/L) for 30 min. After this, the cells were washed with medium 199. The cover slips were then introduced into a gas-tight, temperature-controlled (37 °C), transparent perfusion chamber positioned in the light path of an inverted microscope. Alternating excitation of the fluorescence dye at wavelengths of 340/380 nm was performed with an AR-Cation system adapted to the microscope. Light emitted (500–520 nm) from an area of 10 × 10 µm within a single cell was collected by an ION Optix imaging system. The data are analyzed as the ratio of the light emitted at the 340 to 380 nm wavelength. Cells were stimulated to a field stimulation of 2 Hz.

### 2.6. Western Blot

Isolated cardiomyocytes were incubated with lysis buffer as described before [26]. Samples were loaded on a 15.0% SDS-PAGE and blotted onto membranes. Results were displayed as the ratio of the protein of interest (ornithine decarboxylase, calcium-sensing receptor) normalized to actin as a loading control. 

### 2.7. Statistics

Data are expressed as indicated in the legends. One-way ANOVA and the Student–Newman–Keuls test for post hoc analysis were used to analyze the experiments. Data are presented as box and whisker blots in which the boxes represent the 25, 50, and 75% percentiles, and the whiskers represent the total range of all data. Exact *p* values are given in the legends of the figures. 

## 3. Results

### 3.1. Effect of UA on Load-Free Cell Shortening

As mentioned in the Introduction, UA may directly affect cell function via the activation of protein kinases in a rapid way or, alternatively, via transcriptional effects that would require long-term regulation. Therefore, we analyzed the acute and long-term effects of UA (15 min and 24 h treatment, respectively) on the cell shortening of cardiomyocytes. The long-term incubation of cardiomyocytes with UA decreased cell shortening in a concentration-dependent way, whereas acute exposure had minimal effects (Figure 1A–C). Figure 1D shows a representative cell-shortening record. 

### 3.2. Effect of UA on Arginine Metabolism

As UA effects on the load-free cell shortening of cardiomyocytes required long-term exposure, this favored transcriptional modifications as a mechanistic explanation. Arginine metabolism has multiple connections to UA effects in other cells, as mentioned in the Introduction. Arginine can be metabolized via nitric oxide synthase into nitric oxide and citrulline or via arginase feeding the polyamine metabolism. As moderate increases in nitric oxide improve the load-free cell shortening, whereas UA deteriorated cell shortening, we analyzed the protein expression of ornithine decarboxylase (ODC) as the rate-limiting enzyme of the polyamine metabolism and the protein expression of calcium-sensing receptors. Secreted polyamines may activate this receptor. We found that UA increased ODC expression in cardiomyocytes (Figure 2A,B). In contrast, the expression of the calcium-sensing receptor was not affected (Figure 2A,C).

Next, we inhibited several steps of the polyamine metabolism in the cardiomyocytes. Difluoromethyl ornithine (DFMO) was used to inhibit ODC activity, ornithine was added to stimulate ODC activity, Nor-NOHA was used to inhibit arginase activity, and L-NAME was used to inhibit nitric oxide synthase (NOS) in the cardiomyocytes. DFMO reduced the effect of UA on cell shortening, suggesting that the upregulation of the ODC protein expression contributes to the pathway. Ornithine did not affect the UA effects, and the inhibition of arginase had an intermediate effect between the DFMO and ornithine (Figure 3A–C). L-NAME reduced the cell shortening, but the effect of L-NAME was not further affected by the addition of UA (Figure 3D). The data suggest that the depletion of the NOS substrate (arginine) accelerates arginine consumption via the polyamine metabolism or that L-NAME worsens the function of cardiomyocytes in the presence of UA. The data suggest that arginine depletion may be a common mechanism by which either L-NAME or UA affect cell shortening.

### 3.3. Effect of Extracellular Arginine on UA Effects

The data of the aforementioned experiments support the idea that UA induces an arginine depletion. This may induce a limitation of intracellular arginine bioavailability required for proper function. This hypothesis was tested by increasing the concentration of l-arginine in the medium in the copresence of UA. Increasing the l-arginine concentration, the natural substrate for NOS, attenuated the effect of UA in a concentration-dependent way (Figure 4A,B). However, when l-arginine was replaced by d-arginine, the nonmetabolizable enantiomer of l-arginine, no effect of the arginine supplementation on the UA-dependent loss-of-function was seen (Figure 4C). The experiments supported the hypothesis that high concentrations of UA lead to a reduction in the bioavailability of arginine in cardiomyocytes. 

### 3.4. Effect of Protein Kinase Inhibition on UA Effects

As mentioned in the Introduction, UA may affect the activity of protein kinases. Therefore, we investigated effects of p38 MAPK kinase inhibition (by SB202190), PKC inhibition (by chelerythrine chloride, CEC), and Jun-Kinase inhibition (by SP600125) on the UA-dependent reduction of load-free cell shortening (Figure 5A–C). Among the three signal pathways, only the inhibition of PKC reduced the UA-dependent effect on load-free cell shortening.

### 3.5. Effect of UA on Calcium Sensitization

Reduction in contractility may depend on two different mechanisms that have been reported before. First, the reduced mobilization of calcium from intracellular stores (i.e., sarcoplasmatic reticulum) or, second, the impairment of the calcium sensitivity of troponin subunits. In the first case, calcium transients are decreased, whereas, in the second case, they are normal or even enhanced. To clarify which mechanism may be involved in the UA-dependent effect, we analyzed calcium amplitudes during contraction. As shown in Figure 6A, calcium transients were even improved in the presence of UA despite the attenuating effect on cell shortening. This suggests that the calcium affinity of troponin subunits is altered. If this assumption is correct, the effect of UA should be alleviated by increasing the extracellular calcium concentration or the increased mobilization of intracellular calcium stores via the stimulation of β-adrenoceptors. We tested both possibilities and found that increasing extracellular calcium from 1.25 mmol/L to 1.75 mmol/L completely attenuated the deteriorative effect of UA and the β-agonist isoproterenol mostly attenuated the effect of UA (Figure 6B,C).

### 3.6. Reversibility of UA Effects

The aforementioned data show that cells exposed to UA can be negatively affected by UA with respect to load-free cell shortening. However, it remained unclear whether this effect requires the presence of UA or whether the effect is reversible by removing UA. Therefore, we exposed cultured adult rat ventricular myocytes for 24 h to UA and subsequently replaced the UA-containing medium with normal medium immediately before measuring the cell function. This washing-out period was sufficient to reverse the effect of UA on the load-free cell shortening (Figure 7).

### 3.7. Interaction of Hyperglycemia and UA

Finally, we addressed the question of whether UA synergistically acts with hyperglycemia on load-freed cell shortening because the high plasma levels of UA are often associated with diabetes and hyperglycemia also negatively affects cell shortening. On the cellular basis, hyperglycemia may induce oxidative stress, whereas UA is an ROS scavenger molecule. In this experiment, we confirmed the deteriorating effect of hyperglycemia and UA on cardiomyocytes’ cell shortening. In both cases (glucose and UA concentration), we used a submaximal concentration. Importantly, the effects of both stressors on load-free cell shortening were additive (Figure 8).

## 4. Discussion

The general association between the high plasma levels of UA and heart disease is well established. However, this observation does not answer questions about the mechanism. Is UA a direct stressor for cardiomyocytes, an indicator for oxidative stress in cardiomyocytes, or are the high levels of extracellular UA without intrinsic activity but properly beneficial due to its ROS scavenging character? This study aimed at addressing these questions. The main finding of our study is that UA can directly affect the contractile function of cardiomyocytes. Enzymatic formation of UA by xanthine oxidase can be attenuated by pharmaceuticals, such as allopurinol, and the inhibition of xanthine oxidase activity has beneficial effects in the context of heart disease [27,28]. Nevertheless, even if allopurinol is used, the plasma level of UA is still a predictor of outcome, suggesting a direct effect of UA independent from xanthine oxidase activity in cardiomyocytes [29]. As long as UA levels are still elevated, i.e., by a purine-enriched diet, inhibition of xanthine oxidase will be less sufficient, and this may be explained by the direct deteriorative effects of UA on cardiomyocytes. It was, therefore, important to investigate the effect of UA on cardiomyocytes directly. Our study shows that UA has direct effects on cardiomyocytes. We show that it increases arginine metabolism via the upregulation of ODC, produces functional defects that are in line with the assumption that cells have a functionally relevant arginine deficiency associated with reduced calcium affinity, and which thereby depresses cell function. The data also showed that lowering the concentration of UA will improve function again. The arguments leading to this conclusion will be discussed now in more detail.

The first and novel observation of our study is that the exposure of cardiomyocytes to UA increases the protein expression of ODC. High ODC activity is linked to heart failure as well as an increased polyamine metabolism [30]. Polyamines are required for the growth responses of cells, but they can also be released from cells. In this case, they potentially activate the calcium-sensing receptor in an autocrine manner. However, neither the inhibition of arginase, which directs the arginine metabolism into the direction to polyamine metabolism, nor the inhibition of ODC (with DFMO), nor the activation of ODC (by ornithine), normalizes load-free cell shortening in the presence of UA. In contrast, the inhibition of NOS reduces cell shortening and UA does not further suppress cell shortening. One explanation for this finding may be that NOS activity is required for regular function. Indeed, we already showed years ago that moderate NO formation improves the contractile function of cardiomyocytes [31]. It may be that UA causes a depletion of the cells with arginine, leading to reduction in NOS-dependent nitric oxide formation. UA can affect NOS activity by different pathways: the activation of arginase [32], the inhibition of arginase uptake [13,33], and the inhibition of the interaction between calmodulin and NOS-3 [34]. Each of these steps can lead to NOS uncoupling and increasing oxidative stress [35]. We could suppress the UA-dependent effect by the addition of l-arginine, which can be metabolized, whereas d-arginine does not cause a similar effect. At least in myocardial infarction models, when arginase is activated, the administration of arginine can improve functional recovery again, showing that arginine may become a limited substrate under the stressed condition in the myocardium [36]. Similarly, the administration of l-arginine reverses the detrimental effect of UA in our cell system. Collectively, we suggest that UA by the inhibition of arginine uptake and the direction of the arginine metabolism into the polyamine metabolism, and away from NO-dependent pathways by the induction of ODC, has direct effects on the cell shortening of cardiomyocytes.

We also observed that the effect of UA depends on the activity of the PKC. Interestingly, it was described in other cells that PKC can inhibit the main arginase transport protein in myocytes, CAT-1 [37]. In contrast, the inhibition of other protein kinase signal pathways, such as p38 MAP kinase and c-Jun-kinase, both known to trigger cardiac dysfunction in other contexts, does not modify the responsiveness of the cells to UA. Furthermore, PKC activity is required for ODC induction [38]. UA induces the protein expression of ODC in cardiomyocytes as well. Therefore, UA may trigger various PKC-dependent effects that reduce cell function.

Although arginine depletion may contribute to the UA-dependent effect on cardiomyocytes, the question remains how this translates into the loss of contractility. An earlier study published on myofilaments isolated from diabetic cardiomyocytes showed that PKC-dependent phosphorylation of troponin I is associated with decreased calcium sensitivity [39,40]. Furthermore, propofol decreases myofilament calcium sensitivity in a PKC-dependent way, again showing a coupling of the UA-dependent PKC pathway with UA-dependent effects on calcium affinity, as our study suggests [41]. Our finding that the inhibition of PKC activity attenuates the effect of UA on cell shortening is at least in agreement with the assumption that the PKC-dependent phosphorylation of troponin I reduces calcium sensitivity. In this study, we show, furthermore, that calcium transients are Increased rather than decreased by UA, and that either increasing the extracellular calcium concentration or the activation of β-adrenoceptor stimulation, and thereby increasing the calcium release from internal stores, reverses the effect of UA on cell shortening. Isolated cardiomyocytes can be used as an excellent tool to study direct effects ex vivo. However, the in vivo situation is more complex. Therefore, it might be that high plasma levels of UA that partly impair cardiomyocytes’ function, as shown here, subsequently induces a compensatory activation of the sympathetic nervous systems that then triggers the subsequent stress of cardiomyocytes. Such a crosstalk between the sympathetic activation and UA plasma concentration may also happen in the other direction. There are reports that exercise, which requires sympathetic activation, is associated with the increased plasma concentration of UA [42]. Collectively, our data suggest that UA via PKC activity reduces calcium affinity. The increased calcium transients may be linked to the induction of polyamine metabolism that is nevertheless overridden by calcium desensitization. We suggest that this mechanism will favor the overactivation of the sympathetic nervous system to stabilize cardiac function that then triggers additional stress to the cells.

Our interest to understand the interaction between UA and cardiomyocytes comes also from the diabetes field, as conditions favoring the onset of diabetes, i.e., dietary uptake of large amounts of fructose-rich drinks, increase the plasma concentration of UA. Therefore, high plasma levels of glucose and UA go hand-in-hand. Chickens have naturally high plasma levels of glucose and UA, and it has been shown that this is a double-edged sword [43]. On the one hand, UA reduces oxidative stress caused by hyperglycemia, whereas the combination of high levels of both UA and glucose are leading to reduced function in chicken myocytes [43]. Rat myocytes are normally exposed to lower levels of UA, as rats neither express a renal urate transporter nor uricase, as is found in hominids. However, our study shows a worsening effect of UA on hyperglycemia on isolated cardiomyocytes, even at concentrations that reduce only the moderate effects alone. Hyperglycemia itself induces an ROS-dependent reduction in cell shortening [44]. UA can neutralize ROS formed by cytokines in rat cardiomyocytes but is unable to reduce the effect of hyperglycemia [1]. There is no interaction between both agonists, as one would have expected, due to the antioxidative potential of UA. Therefore, we suggest that the combined effect of UA and hyperglycemia on adult rat cardiomyocytes is an additive effect of two independent mechanisms. In line with this assumption, in the same technical setup as used here (isolated adult rat ventricular myocytes), the negative effect of hyperglycemia can be inhibited by p38 MAP kinase inhibitors [44] but not that of UA (see above). UA induces insulin desensitization in cardiomyocytes, as well [16]. In sum, the data suggest that hyperglycemia in combination with high plasma UA is an independent stress factor in patients with metabolic syndrome.

## 5. Study Limitations

As with all studies, this study has some limitations. First, the UA concentration used here leans on plasma concentrations typically found in humans, but rat cardiomyocytes are normally exposed to lower concentrations of UA. Second, the cellular arginine pool and arginine turnover were not biochemically quantified here, and the argument that arginine depletion triggers the UA-dependent effect comes from experiments showing the dependency of the effects from extracellular arginine. Third, as mentioned already, the observed stress by high UA levels may lead to compensatory mechanisms in vivo, such as the activation of the sympathetic nervous system shifting the disease progression to other receptor systems. Finally, load-free cell shortening is used as a well-established surrogate parameter of cardiomyocytes’ function, but in vivo UA may also affect perfusion and imbalance the energy supplementation of myocytes. Therefore, the mechanistic insights outworked in this study require proper in vivo controls. Nevertheless, these limitations are opposed to the advantages of the mechanistic analysis with terminally differentiated cardiomyocytes and the impressive high reproducibility of the effects.

## 6. Conclusions

Our study used the advantage of an isolated cell model to show that UA has direct effects on cardiomyocytes. This effect adds stress to the heart. On the molecular basis, UA effects are linked to the arginine metabolism (specifically the acceleration of the polyamine metabolism and arginine uptake, leading to intracellular arginine depletion) and calcium desensitization most likely triggered by this process. The conclusions drawn from our cell-based experimental study are summarized in Figure 9.

## Figures and Tables

**Figure 1 biology-12-00004-f001:**
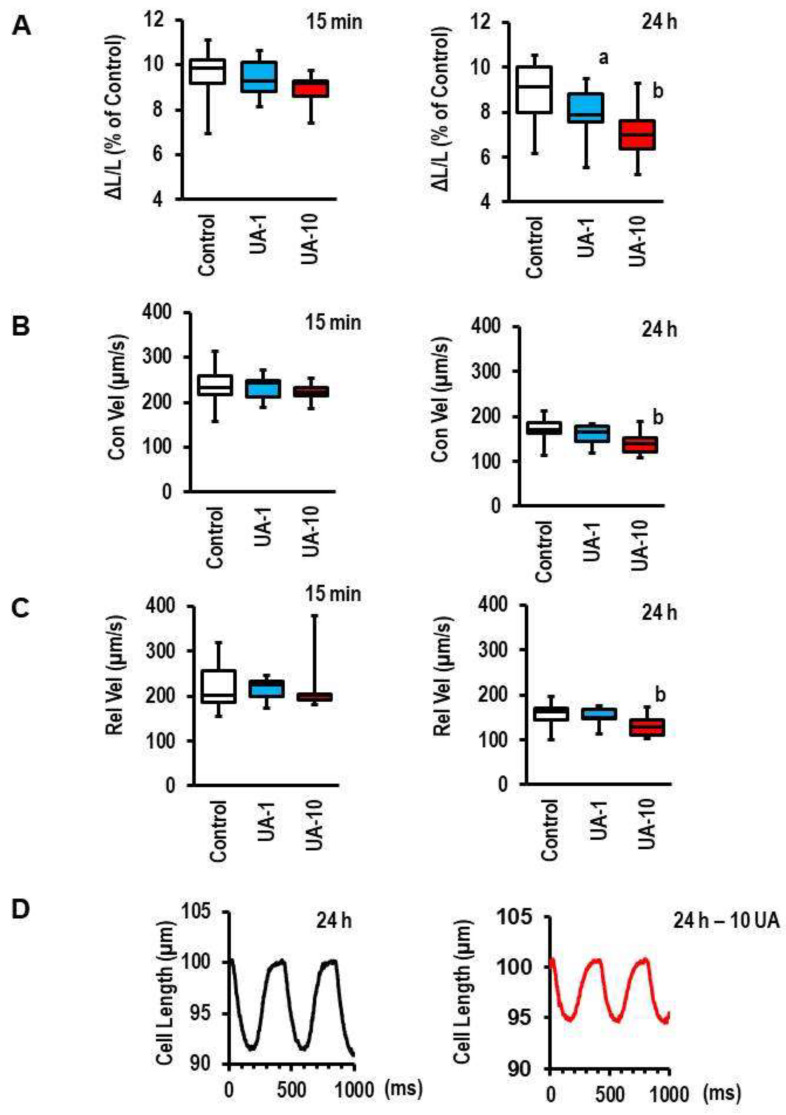
Effect of uric acid (UA) on cell shortening of cardiomyocytes. Cells were treated with UA for 15 min or 24 h, as indicated. Concentrations used were 1 mg/dL (UA-1) or 10 mg/dL (UA-10). Data show results for relative cell shortening (shortening amplitude [ΔL] normalized to diastolic cell length [L] and expressed as percent cell shortening (**A**), contraction velocity (Con-Vel, (**B**)) expressed in µm per s, and relaxation velocity (Rel-Vel, (**C**)) expressed in µm per s. Representative single-cell recordings for 24 h UA exposure (10 mg/dL) are given in (**D**). Data are means from n = 11 (control, 10 min), n = 10 (UA-1, 10 min), n = 6 (UA-10, 10 min), n = 24 (control, 24 h), n = 13 (UA-1, 24 h), and n = 22 (UA-10, 24 h) preparations (rats), with 4 cell culture dishes per preparation and 9 cells evaluated per cell-culture dish. Statistical analysis is based on preparations. Data are expressed as bars and whiskers with median, 25% quartile, 75% quartile indicated by bars, and complete range by whiskers. Statistical analysis is based on one-way ANOVA and Student–Newman–Keuls post hoc analysis. a, *p* < 0.05 vs. control and UA-10; b, *p* < 0.05 vs. control and UA-1.

**Figure 2 biology-12-00004-f002:**
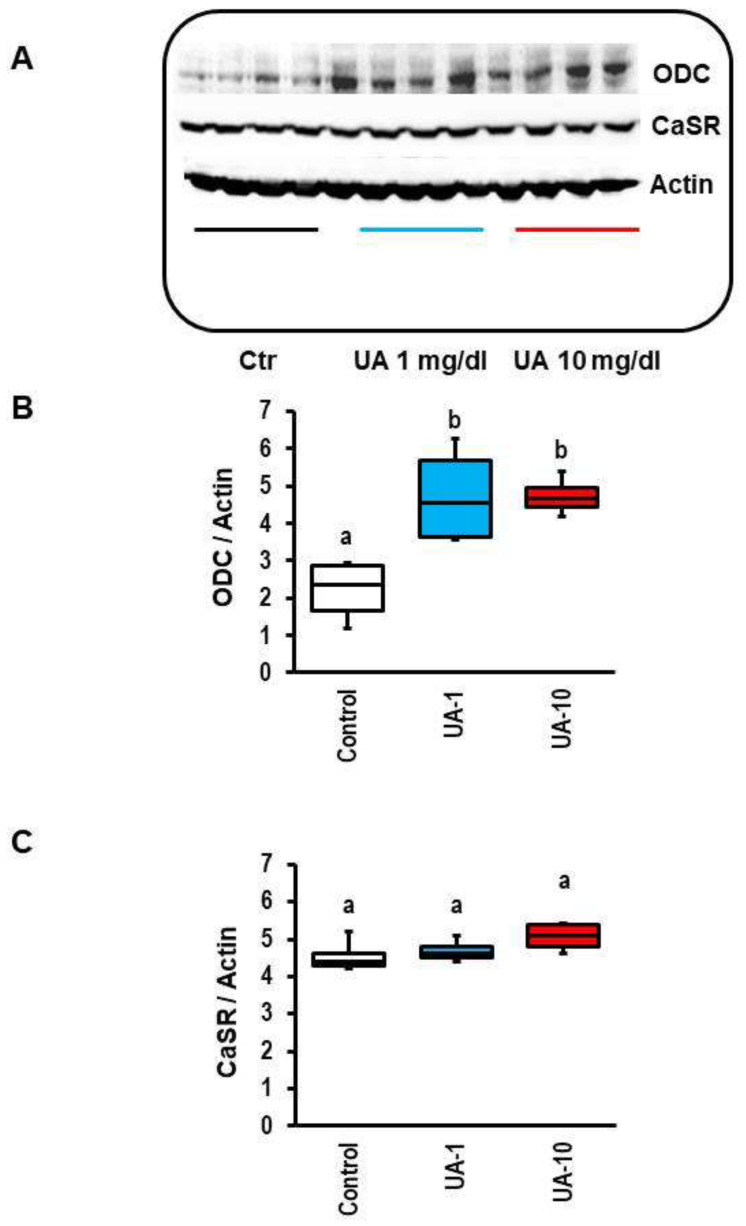
Effect of 24 h uric acid (UA) exposure on ornithine decarboxylase (ODC) and calcium-sensing-receptor (CaSR) expression. (**A**) Representative Western blot indicating the expression of ODC, CaSR, and Actin, which was used for normalization. (**B**,**C**) Quantitative analysis of ODC and CaSR expression, respectively. Data are expressed as bars and whiskers with median, 25% quartile, 75% quartile indicated by bars, and complete range by whiskers. Statistical analysis is based on one-way ANOVA and Student–Newman–Keuls post hoc analysis. a, *p* < 0.05 vs. control. Each n = 4 as indicated in (**A**). Equal letters (a, b) indicated sample groups with *p* > 0.05. Different letters indicate *p* < 0.05.

**Figure 3 biology-12-00004-f003:**
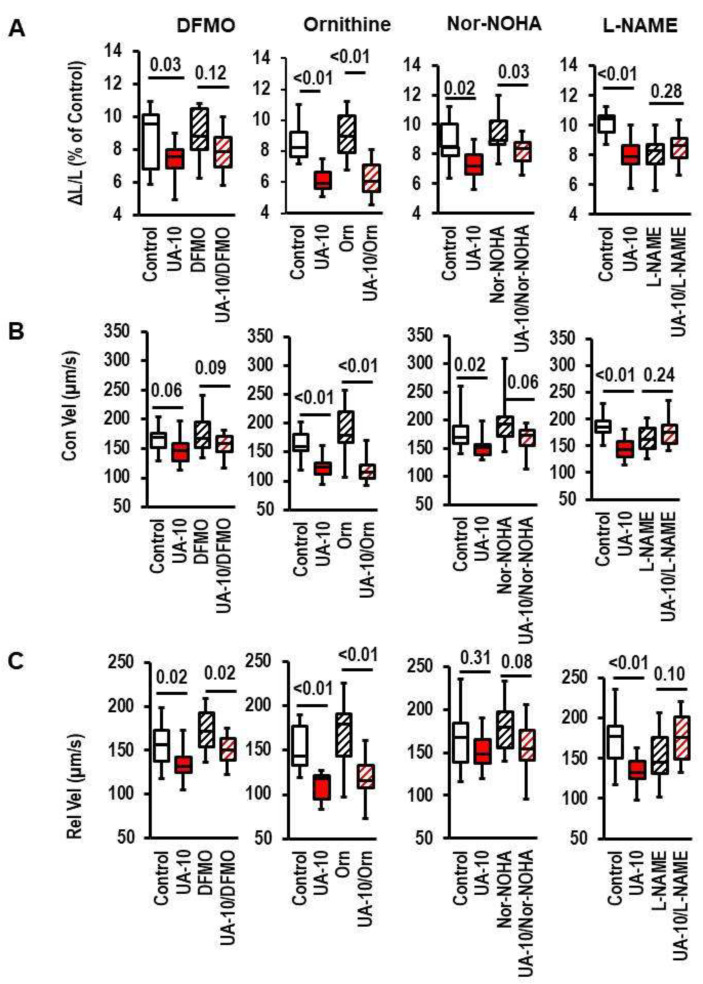
Effect of difluoromethyl ornithine (DFMO), ornithine (Orn), Nω-Hydoxy-l-arginine-acetate (Nor-NOHA), or L-N^G^-Nitro arginine methyl ester (L-NAME) on cell shortening and uric-acid (UA)-dependent effects. Data show the effects on cell shortening ((**A**); ΔL/L), contraction velocity ((**B**); Con Vel), and relaxation velocity ((**C**); Rel-Vel). Data are expressed as bars and whiskers with median, 25% quartile, 75% quartile indicated by bars, and complete range by whiskers. Statistical analysis is based on Student *t*-tests and exact *p*-values are indicated. Data are means from n = 10 (each) preparations (rats), with 4 cell culture dish per preparation and 9 cells evaluated per cell-culture dish. Statistical analysis is based on preparations. Two-way ANOVA was performed to exclude an interaction between the UA effects and that of the inhibitors.

**Figure 4 biology-12-00004-f004:**
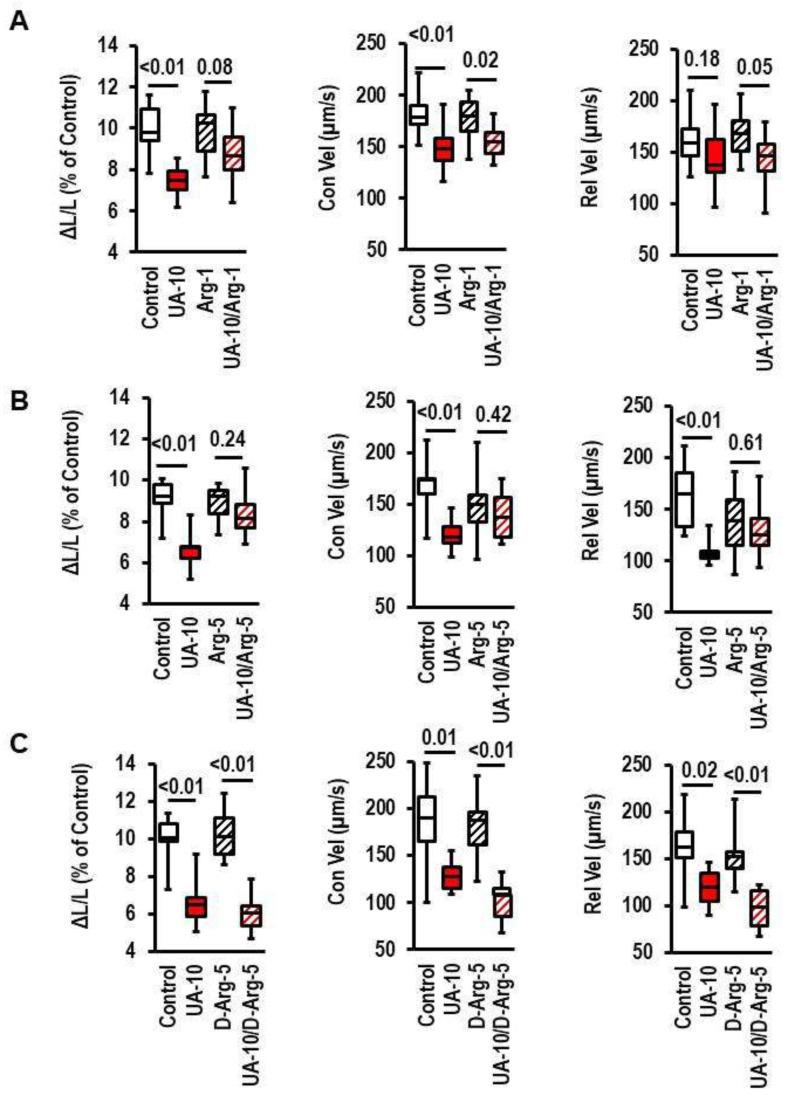
Effect of arginine supplementation on cell shortening and uric-acid (UA)-induced effects. Cells were exposed to L-arginine (Arg-1; 1 µmol/L; Arg-5, 5 µmol/L) or D-arginine (D-Arg, 5 µmol/L) and UA-10 (10 mg/dL). Data show the effects on cell shortening ((**A**); ΔL/L), contraction velocity ((**B**); Con Vel), and relaxation velocity ((**C**); Rel-Vel). Data are expressed as bars and whiskers with median, 25% quartile, 75% quartile indicated by bars, and complete range by whiskers. Statistical analysis is based on Student *t*-tests and exact *p*-values are indicated. Data are means from n = 10 (each) preparations (rats), with 4 cell culture dish per preparation and 9 cells evaluated per cell culture dish. Statistical analysis is based on preparations.

**Figure 5 biology-12-00004-f005:**
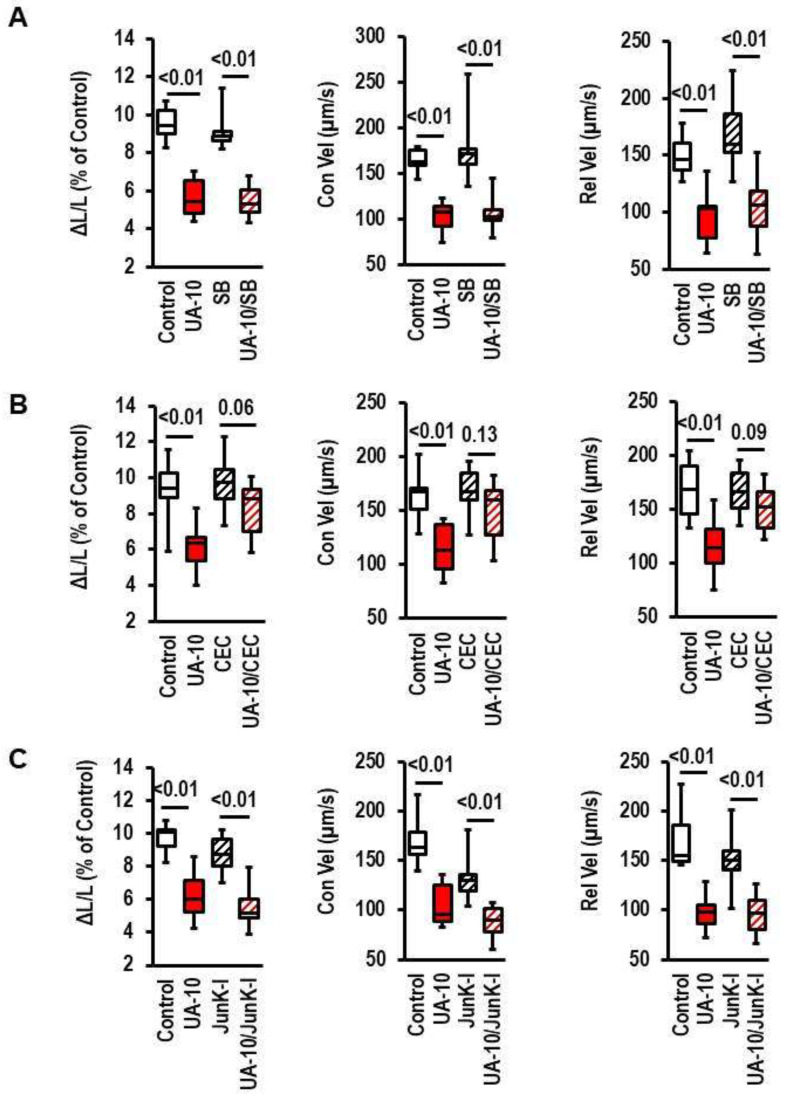
Effect of inhibition of signal transduction pathways on cell shortening and uric-acid (UA)-induced effects. Cells were exposed to SB202190 (SB; 10 µmol/L), chelerythrine chloride (CEC, 10 µmol/L), or SP600125 (SP, 10 µmol/L) and UA-10 (10 mg/dL). Data show the effects on cell shortening ((**A**); ΔL/L), contraction velocity ((**B**); Con Vel), and relaxation velocity ((**C**); Rel-Vel). Data are expressed as bars and whiskers with median, 25% quartile, 75% quartile indicated by bars, and complete range by whiskers. Statistical analysis is based on Student *t*-tests and exact *p*-values are indicated. Data are means from n = 10 (each) preparations (rats), with 4 cell culture dish per preparation and 9 cells evaluated per cell culture dish. Statistical analysis is based on preparations.

**Figure 6 biology-12-00004-f006:**
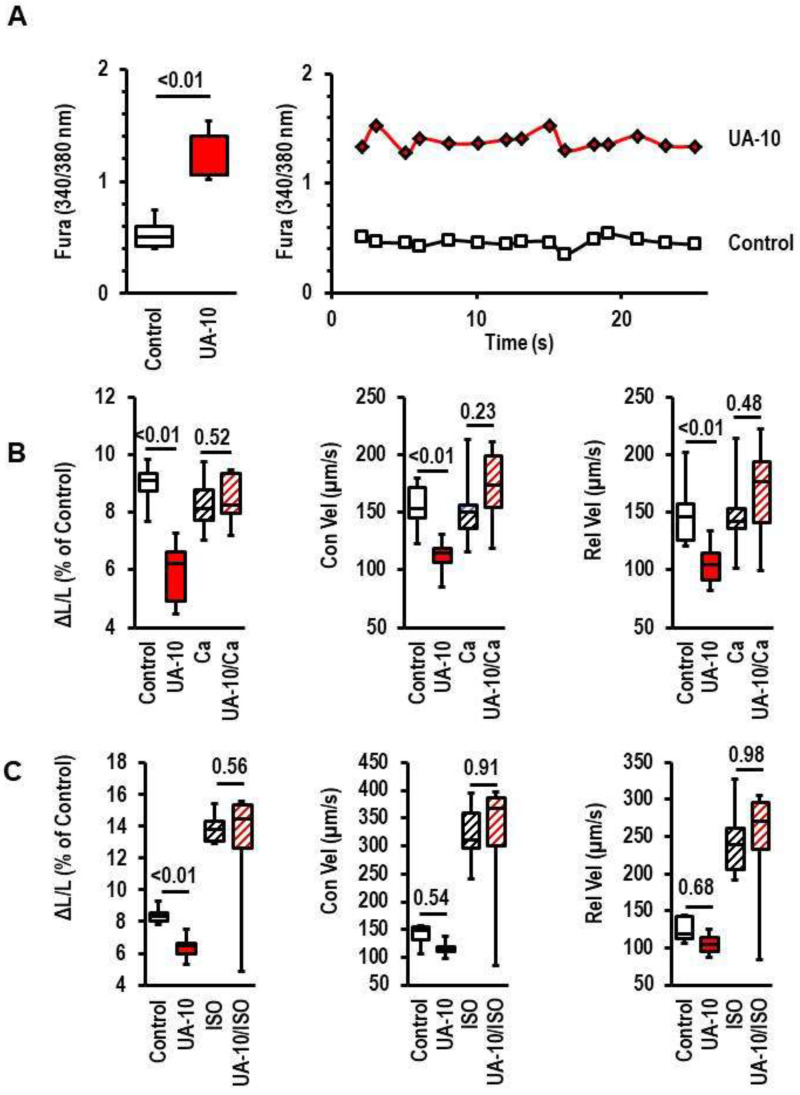
Effect of uric acid (UA) on calcium transients (**A**) and of calcium mobilization on UA-induced effects on cell shortening. In (**B**), extracellular calcium concentration is increased from 1.25 mmol/L (control) to 1.75 mmol/L (Ca). In (**C**), the β-adrenoceptor agonist isoprenaline (ISO, 10 nmol/L) was added. Data are expressed as bars and whiskers with median, 25% quartile, 75% quartile indicated by bars, and complete range by whiskers. Statistical analysis is based on Student *t*-tests and exact *p*-values are indicated. Data show the effects on cell shortening ((**A**); ΔL/L), contraction velocity ((**B**); Con Vel), and relaxation velocity ((**C**); Rel-Vel). Data are means from n = 5 (calcium transients) or n = 10 (**B**,**C**) preparations (rats), with 4 cell culture dish per preparation and 9 cells evaluated per cell culture dish. Statistical analysis is based on preparations. In (**A**), a representative single-cell recording for two cells is also given for 15 consecutive calcium transients.

**Figure 7 biology-12-00004-f007:**
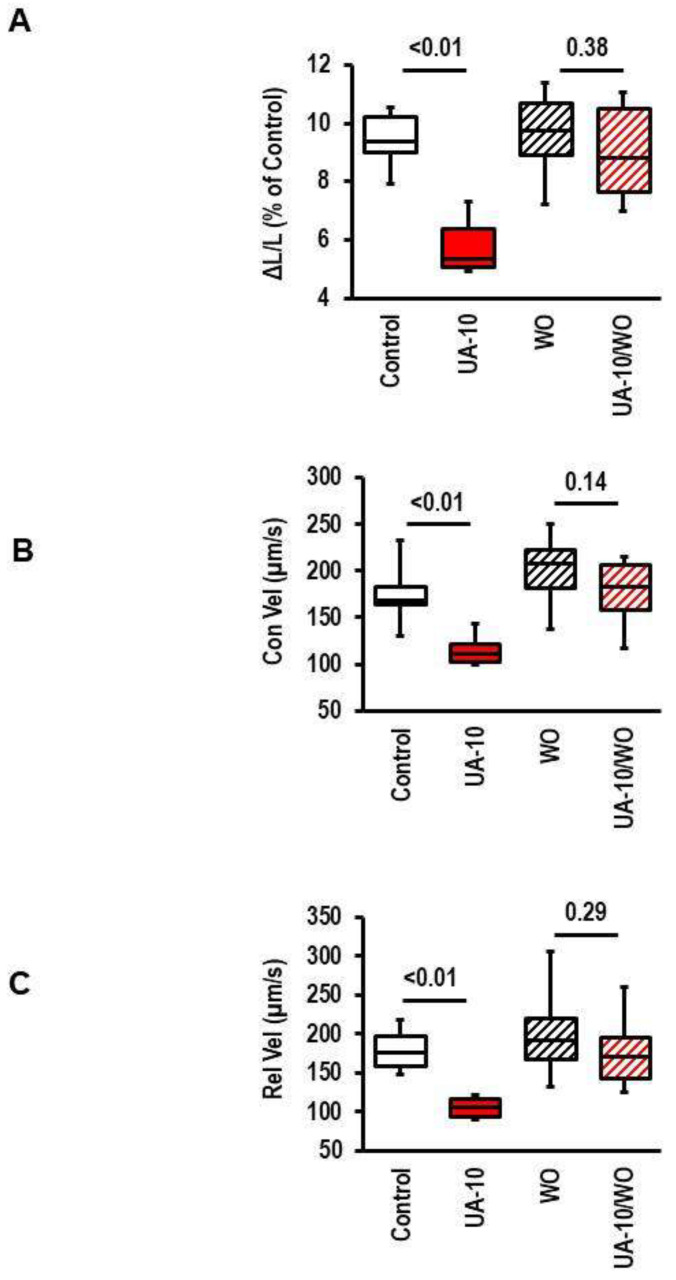
Effect of uric acid (UA) withdrawn for cell shortening measurements. Cells were exposed to UA for 24 h and cell shortening was analyzed in the presence of UA or after 15 min of the withdrawing of UA before starting to measure the cell shortening (WO). Data show the effects on cell shortening ((**A**); ΔL/L), contraction velocity ((**B**); Con Vel), and relaxation velocity ((**C**); Rel-Vel). Data are expressed as bars and whiskers with median, 25% quartile, 75% quartile indicated by bars, and complete range by whiskers. Statistical analysis is based on Student *t*-tests and exact *p*-values are indicated. Data are means from n = 10 (each) preparations (rats), with 4 cell culture dish per preparation and 9 cells evaluated per cell culture dish. Statistical analysis is based on preparations.

**Figure 8 biology-12-00004-f008:**
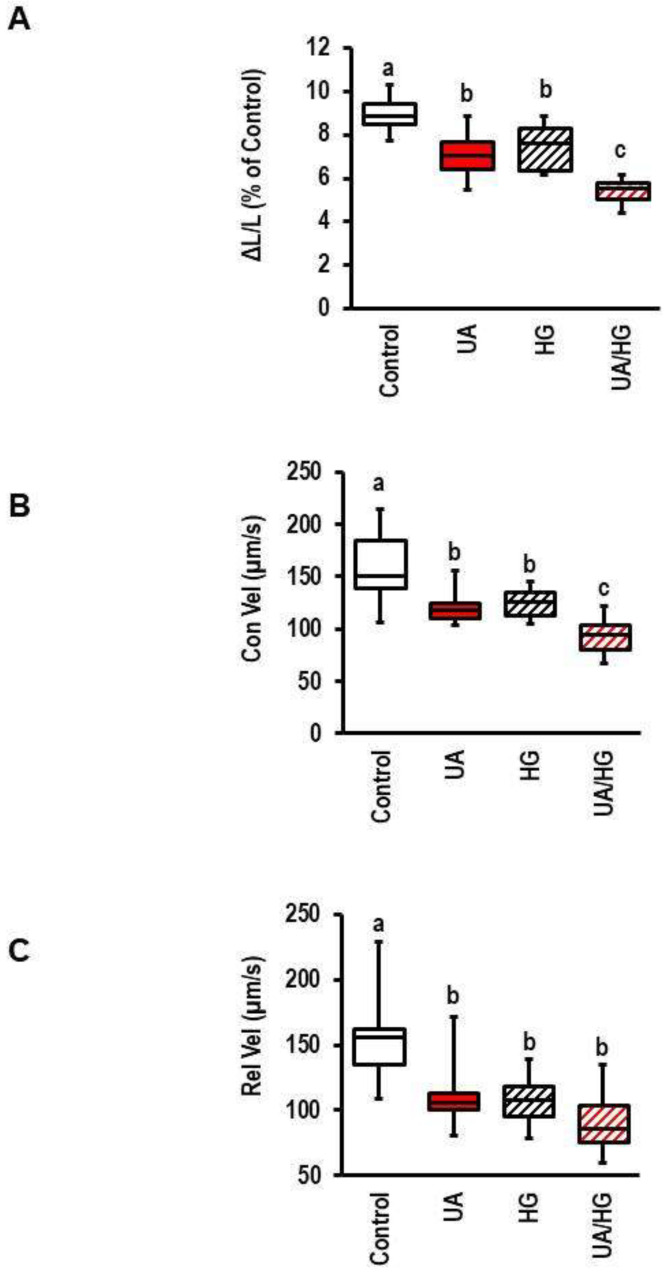
Effect of hyperglycemia on uric-acid (UA)-induced effects on cell shortening. Cells were exposed to UA (3 mg/dL) and glucose (15 mmol/L; HG) for 24 h before analyzing cell shortening. Data show the effects on cell shortening ((**A**); ΔL/L), contraction velocity ((**B**); Con Vel), and relaxation velocity ((**C**); Rel-Vel). Data are expressed as bars and whiskers with median, 25% quartile, 75% quartile indicated by bars, and complete range by whiskers. Statistical analysis is based on Student *t*-tests and exact *p*-values are indicated. Data are means from n = 10 (each) preparations (rats), with 4 cell culture dish per preparation and 9 cells evaluated per cell culture dish. Statistical analysis is based on preparations. Equal letters (a, b) indicate sample groups with *p* > 0.05. Different letters indicate *p* < 0.05.

**Figure 9 biology-12-00004-f009:**
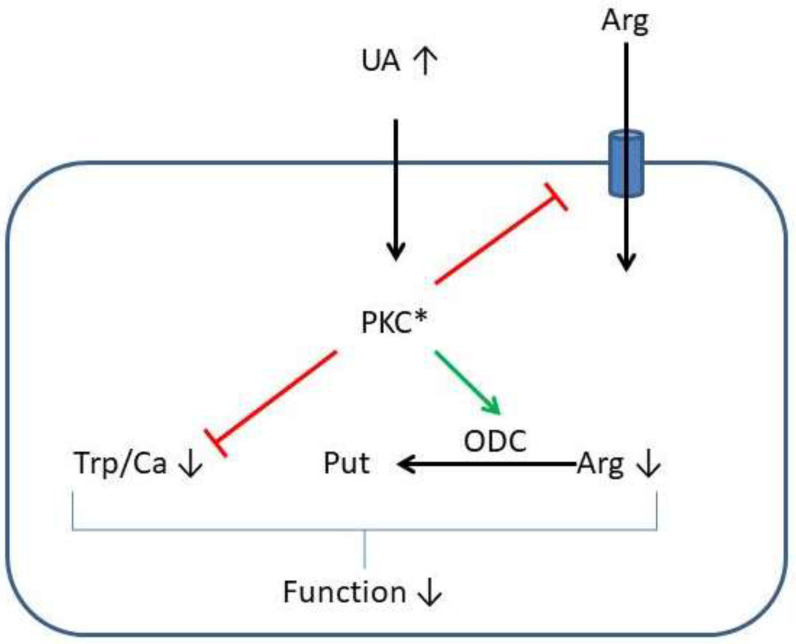
Summary of the findings of this study and schematic overview how these data may explain the long-term effect of high concentrations of UA on the cell shortening (function) of cardiomyocytes (see Figure 1). High concentrations of extracellular UA (UA ↑) activate PKC-dependent pathways (PKC*), as suggested from the experiments with a PKC-inhibitor (Figure 5B). Subsequently, arginine uptake is affected (see Figure 4), ODC is induced (see Figure 2), and the arginine metabolism altered (Figure 3 and Figure 4). Function is impaired in the presence of high intracellular calcium (see Figure 6).

## Data Availability

Original data can be requested by the corresponding author.

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
