# Peer review of "Uric Acid Deteriorates Load-Free Cell Shortening of Cultured Adult Rat Ventricular Cardiomyocytes via Stimulation of Arginine Turnover"

_biology, 2022, doi:10.3390/biology12010004_

Round 1

Reviewer 1 Report

The article 'Uric acid deteriorates load free cell shortening of cultured adult rat ventricular cardiomyocytes via stimulation of arginine turnover' by Weber et al. describes the effect of plasma-relevant concentrations of uric acid on adult rat ventricular cardiomyocytes. The results presented here are mostly convincing, informative, and relevant to the readership of 'Biology.' However, the authors need to address some critical concerns in the current version of the manuscript before its publication. The issues are listed below.

How did the authors choose 15 min and 24 h treatments of UA on cardiomyocytes? 

Did the authors observe any effect of UA treatment on gross cardiomyocyte size or myocardial hypertrophy?

Have the authors ruled out any apoptosis in cardiomyocytes at the given UA concentrations? 

It is not immediately clear what the color bars in Figure 2A represent. Do they represent replicates? If so, the expression does not look similar. 

The rationale behind testing the effect of UA on PKC is underdeveloped. Please elaborate in section 3.4. 

Can the authors comment on the ROS generation upon UA treatment for 15 min and 24 h? 

It would be helpful for the readers to provide a model for the effect of UA treatment on cardiomyocytes.   

There are many verb tense inconsistencies throughout the manuscript. Further, the writing style, typographical and grammatical errors should be corrected in the revised version of the manuscript.

Author Response

You wrote: The article 'Uric acid deteriorates load free cell shortening of cultured adult rat ventricular cardiomyocytes via stimulation of arginine turnover' by Weber et al. describes the effect of plasma-relevant concentrations of uric acid on adult rat ventricular cardiomyocytes. The results presented here are mostly convincing, informative, and relevant to the readership of 'Biology.' However, the authors need to address some critical concerns in the current version of the manuscript before its publication. The issues are listed below.

Response: We thank the reviewer for his/her helpful comments. Please find enclosed our point-by-point responses.

You wrote: How did the authors choose 15 min and 24 h treatments of UA on cardiomyocytes?

Response: The authors use the model for many years and have a great experience with the cultivation and stimulation of adult rat ventricular myocytes. We use myocytes either in acute experiments which aimed at analysing any direct effects mostly linked to alteration in calcium transients or in long-term experiments aimed at analysing any effects that are potentially affected by transcription. Acute experiments are performed within 15 min of exposure (as done in this study) and long-term experiments are performed after 24 h of exposure (as also done in this study). Further extension of incubation times is difficult as the myocytes are exposed to serum-free media for such experiments. They are quiescent (non-beating) cells. Therefore, they are not contracting during exposure time and thereby decrease their contractile proteins (see Cardiomyocytes – Active Players in Cardiac Disease; ed. Klaus-Dieter Schlüter; Chapter 1, Table 1.3; Page 15; 2016 (doi: 10.1007/978-3-319-31251-4). Under such conditions cells are more or less stable for 24-48 h. Examples in which we established the protocol for acute effects and performed similar experiments are given in doi: 10.1007/s00424-014-1498-y (Gadolinium, Putrescine); doi: 10.1210/en.2004-1180 (TIP39); doi: 10.1016/j.yjmcc.2006.10.012 (NO). Examples in which we established the protocol for long-term experiments are given in doi: 10.1002/jpc.24613 (Effect of IL-6); doi: 10.1007/s00424-014-1498-y (Calcium Sensor); doi: 10.1002/pcp.25612 (Effect of endothelin-1); doi: 10.1007/s00395-017-0650-1 (oxLDL); doi: 10.1007/s003595-020-00824-w (PCSK9). In this study we used therefore these previously established conditions.     

You wrote: Did the authors observe any effect of UA treatment on gross cardiomyocyte size or myocardial hypertrophy?

Response: No, we did not observe any effect on cell sizes. I.e., the mean diastolic cell lengths were 106 µm (control), 107 µm (0.1 mg/dl uric acid; p=0.47); 106 µm (1.0 mg/dl uric acid; p=0.95) and 109 µm (10.0 mg/dl uric acid; p=0.16) (n=125 cells each). There were neither differences between groups that reached the level of significance nor a clear concentration-response-relationship. Therefore, we have no evidence for hypertrophy. 

You wrote: Have the authors ruled out any apoptosis in cardiomyocytes at the given UA concentrations? 

Response: Based on the gross morphology of the cells and the equal number of cells that responded to electrical stimulation, we did not have any evidence for necrotic or apoptotic events and did not further proceed with any analysis into this direction.

You wrote: It is not immediately clear what the color bars in Figure 2A represent. Do they represent replicates? If so, the expression does not look similar. 

Response: The blot shows n=4 for control (lane 1-4, black), UA 1 mg/dl (lane 5-8, blue), and UA 10 mg/dl (lane 9-12; red). Same colours are used for description of quantification as shown in 2B. We disagree with the reviewers comment concerning variations. Some variations are clearly indicated in the quantification seen in 2B by the range but did not affect the conclusion. Yes, the band for ODC in lane 3 is stronger than in lane 1,2, and 4, but it is realistic to expect some biological variations. We still believe that the blot really shows an increased expression of OCD but not of CaSR in UA-treated cells cultures. This conclusion was subsequently validated by mechanistic experiments shown in Fig. 3.

You wrote: The rationale behind testing the effect of UA on PKC is underdeveloped. Please elaborate in section 3.4. 

Response: The rational to investigate the effect of PKC on UA treatment is first explained in the introduction. Please read: “In this context it is interesting that UA itself can affect cell function directly. UA can inhibit arginine uptake in endothelial cells [13], activate mitogen activated protein kinase (MAPK) pathways in pancreatic cells [14], increases the expression of voltage-dependent potassium channels in atrial myocytes [15], and induces insulin resistance in cardiomyocytes [16], to name a few of cell-specific effects of UA. Furthermore, it was reported that UA inhibits arginine uptake of cells via inhibition of the main transporter, CAT1, in a protein kinase C (PKC)-dependent way [13].”

We also discussed this topic in a broad way in the discussion as the result of this experiment led us exploring the effect on calcium transients. Please read: “We also observed that the effect of UA depends on the activity of PKC. Interestingly, it was described in other cells that UA can activate PKC and the main arginase transport protein in myocytes, CAT-1, s inhibited in a PKC-dependent way [13,37]. In contrast inhibition of other protein kinase signal pathways, such as p38 MAP kinase and c-Jun-kinase, both known to trigger cardiac dysfunction in other contexts, did not modify the responsiveness of the cells to UA. Furthermore, PKC activity is required for ODC induction [37]. UA induced the protein expression of ODC in cardiomyocytes as well. Therefore, OA may trigger various PKC-dependent effects that reduce cell function.”

As the result section should focus on the description of the experiments we did not further elaborate this topic in the result section. We used the following introduction sentence because at that time it was not clear whether PKC-inhibition affects the signal and whether other protein kinases may also interfere. Please read: “As mentioned in the introduction, UA may affect the activity of protein kinases.”  

You wrote: Can the authors comment on the ROS generation upon UA treatment for 15 min and 24 h?

Response:  We discussed the relationship between UA and ROS in the introduction. Please read: “UA itself has antioxidative properties and is has been suggested that high plasma concentrations of UA in hominids contributes to its longevity compared to small rodents [1-5]. In line with this, mice that were haplo-deficit for uricase had an increased longevity [5].” Based on this knowledge and in combination that plasma-relevant concentrations of UA did not damage the cells except of reducing the cell shortening amplitude (see discussion before), we did not measure ROS formation in these cells. The observed induction of ODC shifts arginine metabolism into the direction of polyamines instead of nitric oxide but polyamines themselves do not lead to oxidative stress in this system. In contrast, oxLDL in the same system led to p38 MAP kinase-dependent oxidative stress (doi: 10.1007/s00395-017-0650-1). However, we excluded in our study a participation of p38 MAP kinase (see Fig. 5). Therefore, we have no indication that ROS plays any role in the observed effects.     

You wrote: It would be helpful for the readers to provide a model for the effect of UA treatment on cardiomyocytes.  

Response: We agree with your comment and have added such a model (see Fig. 9). 

You wrote: There are many verb tense inconsistencies throughout the manuscript. Further, the writing style, typographical and grammatical errors should be corrected in the revised version of the manuscript.

Response: We thank you for your advice and checked carefully the manuscript.

Reviewer 2 Report

Congratulations for this excelent article. 

Author Response

You wrote: Congratulations for this excelent article. 

Response: We thank the reviewer for his/her time spent by evaluating our manuscript.

Reviewer 3 Report

In the manuscript, the authors reported Uric acid (UA) deteriorates load free cell shortening of cultured adult rat ventricular cardiomyocytes via stimulation of arginine turnover. They found that UA has direct effects on cardiomyocytes. This effect effects are linked to arginine metabolism (specifically acceleration of polyamine metabolism and arginine uptake leading to intracellular arginine depletion). The manuscript is well organized and could be accepted after following minor revision.

1.    In Figure 1, why 15 min and 24 h treatment could be determined as the acute and long-term effect of UA, the author should explain it.

2.    What’s the difference between this manuscript and reference 13?

3.    It is recommended to show the P-value results of fig. 2C to illustrate that UA did not affect calcium-sensing receptor expression in cardiomyocytes. The P-value of fig.8 is recommended to be marked in the figure or explained in the notes;

4.    Please correct typos and carefully check all the manuscript.

5.    The references should be modified according to the target journal. Please check the relevant contents carefully for any omissions.

6.    It is suggested to add a scheme about the mechanism of this manuscript.

Author Response

You wrote: In the manuscript, the authors reported Uric acid (UA) deteriorates load free cell shortening of cultured adult rat ventricular cardiomyocytes via stimulation of arginine turnover. They found that UA has direct effects on cardiomyocytes. This effect effects are linked to arginine metabolism (specifically acceleration of polyamine metabolism and arginine uptake leading to intracellular arginine depletion). The manuscript is well organized and could be accepted after following minor revision.

Response: We thank the reviewer for his/her helpful comments. Please find enclosed our point-by-point responses. 

You wrote: 1.    In Figure 1, why 15 min and 24 h treatment could be determined as the acute and long-term effect of UA, the author should explain it.

Response: The authors use the model for many years and have a great experience with the cultivation and stimulation of adult rat ventricular myocytes. We use myocytes either in acute experiments which aimed at analysing any direct effects mostly linked to alteration in calcium transients or in long-term experiments aimed at analysing any effects that are potentially affected by transcription. Acute experiments are performed within 15 min of exposure (as done in this study) and long-term experiments are performed after 24 h of exposure (as also done in this study). Further extension of incubation times is difficult as the myocytes are exposed to serum-free media for such experiments. They are quiescent (non-beating) cells. Therefore, they are not contracting during exposure time and thereby decrease their contractile proteins (see Cardiomyocytes – Active Players in Cardiac Disease; ed. Klaus-Dieter Schlüter; Chapter 1, Table 1.3; Page 15; 2016 (doi: 10.1007/978-3-319-31251-4). Under such conditions cells are more or less stable for 24-48 h. Examples in which we established the protocol for acute effects and performed similar experiments are given in doi: 10.1007/s00424-014-1498-y (Gadolinium, Putrescine); doi: 10.1210/en.2004-1180 (TIP39); doi: 10.1016/j.yjmcc.2006.10.012 (NO). Examples in which we established the protocol for long-term experiments are given in doi: 10.1002/jpc.24613 (Effect of IL-6); doi: 10.1007/s00424-014-1498-y (Calcium Sensor); doi: 10.1002/pcp.25612 (Effect of endothelin-1); doi: 10.1007/s00395-017-0650-1 (oxLDL); doi: 10.1007/s003595-020-00824-w (PCSK9). In this study we used therefore these previously established conditions.      

You wrote: 2.    What’s the difference between this manuscript and reference 13?

Response: We thank the reviewer for his/her comment. We now describe the previous finding in a more correct way (see ref. 13/37).  There are two important differences between previous studies and our study: First, in study 13/37 rat aortae were used to study the link between PKC and arginine metabolism whereas we used adult rat ventricular cardiomyocytes and suggest a similar mechanism on cardiomyocytes. Second, in study 13/37 is no link to uric acid as done in our study as increased PCK activity is related to ageing but not UA. The link to UA and PKC is given in ref. 13. Therefore, we newly couple PKC activation with arginine metabolism and this in a completely new context (Cardiomyocytes vs. vascular cells).   

You wrote: 3.    It is recommended to show the P-value results of fig. 2C to illustrate that UA did not affect calcium-sensing receptor expression in cardiomyocytes. The P-value of fig.8 is recommended to be marked in the figure or explained in the notes;

Response: We thank the reviewer for this remark and have corrected Fig. 2B, 2C and the legend of Fig.8.

You wrote: 4.    Please correct typos and carefully check all the manuscript.

Response: We thank you for your advice and checked carefully the manuscript.

You wrote: 5.    The references should be modified according to the target journal. Please check the relevant contents carefully for any omissions.

Response: We thank you for your advice and checked carefully the manuscript.

You wrote: 6.    It is suggested to add a scheme about the mechanism of this manuscript.

Response: We agree with your comment and have added such a model. 

Round 2

Reviewer 1 Report

The authors have addressed all the issues raised in the previous version of the manuscript. Hence, the paper can now be accepted for publication.